# Design and Experiment of an Ultrasound-Assisted Corneal Trephination System

**DOI:** 10.3390/mi14020438

**Published:** 2023-02-12

**Authors:** Jingjing Xiao, Jialong Chen, Mengqiong Li, Leiyu Zhang

**Affiliations:** 1College of Computer and Information Engineering, Xiamen University of Technology, Xiamen 361024, China; 2Affiliated Xiamen Eye Center, Xiamen University, Xiamen 361001, China; 3Beijing Key Laboratory of Advanced Manufacturing Technology, Beijing University of Technology, Beijing 100124, China

**Keywords:** corneal trephination, lamellar keratoplasty, ultrasound-assisted, eccentric adjustment

## Abstract

According to the advantages of ultrasonic vibration cutting, an ultrasound-assisted corneal trepanation robotic system is developed to improve the accuracy of corneal trephination depth and corneal incision quality in corneal trephination operations. Firstly, we analyzed the reasons for the difficulty in controlling the depth of trephination in corneal transplantations from the perspective of the biomechanical properties of the cornea. Based on the advantages of ultrasonic vibration cutting, we introduced an ultrasonic-vibration-assisted cutting method for corneal trephination and analyzed the cutting mechanism. Secondly, we described the surgical demands of corneal trephination and listed the design requirements of a robotic system. Thirdly, we introduced the design details of said system, including the system’s overall structure, the ultrasound-assisted end effector, the key mechanisms of the robotic system, and the human–machine interaction interface. We designed the end effector based on ultrasonic vibration cutting and its eccentric adjustment system in an innovative way. Additionally, we then presented a procedure for robot-assisted corneal trephination. Finally, we performed several cutting experiments on grapes and porcine eyeballs in vitro. The results show that, compared with manual trephine, ultrasound-assisted corneal trephination has a better operation effect on the accuracy of corneal trephination depth and corneal incision quality.

## 1. Introduction

Corneal transplantations are carried out to replace turbid and diseased corneal tissue with normal transparent corneal tissue, being one of the most important operations in ophthalmology [1]. The first successful human corneal transplantation was performed in Germany in 1886 by von Hippel, who transplanted a full-thickness rabbit cornea onto a young girl’s eye, restoring her vision [2]. Von Hippel developed the first corneal trephine, which was motorized (wind-up) and became the prototype for future devices. Currently, the most common corneal transplantations are penetrating keratoplasty (PK) and deep anterior lamellar keratoplasty (DALK) [3]. The former is one of the most successful forms of tissue transplantation, and it is the preferred surgical form for full-layer corneal infections and trauma, full-layer turbidity, or white spots [4]. When keratopathies do not affect the endothelium and Descemet’s membrane (DM) the latter can replace the former in the treatment of lesions above the deep stromal layer of the cornea to reduce the risk of immune rejection [3,5]. Paufique noted that lamellar keratoplasty was technically more difficult than penetrating keratoplasty [6]. These difficulties are mainly represented in the difficult maneuver of lamellar corneal dissection and the difficulty in controlling the depth of corneal trephination. In addition, some researchers analyzed the biomechanical properties of transplanted corneas after PK and DALK, and they found that some healing responses may occur at the recipient donor interface, which can influence the biomechanical properties of a graft as a whole [4,7]. The healing effect of corneal transplantation is closely related to the morphology of a corneal cutting surface. The less cell loss during trephination the better the morphology of a corneal cutting surface. Improved surgical equipment, instruments, and techniques may minimize this loss. Emerging technologies and applications, such as femtosecond lasers and robotic trephinations, have developed rapidly [8,9,10]; however, as femtosecond laser cutting requires negative pressure suction, intraocular pressure will increase. Consequently, corneas on the verge of perforation and other laser contraindications are not suitable for femtosecond laser trephination [11]. As for robotic trephination, although it has a better trephination effect than manual trephination, it cannot control trephine depth in DALK precisely due to the hyperelasticity and viscoelasticity of corneas [9,12].

In recent years, ultrasound has been widely used in the biomedical field, which can be divided into ultrasonic therapy and ultrasonic diagnosis equipment. Ultrasonic scalpels are an important representation of ultrasonic therapy equipment. As a major part of surgical equipment, ultrasonic scalpels have become a research hotspot in related fields at home and abroad because of their characteristics of high cutting precision, less bleeding, little smoke, and fast postoperative recovery [13]. With the development of modern biomedicine and the progress in medical technology, ultrasonic micromanipulation tools have also been introduced to micro-operating objects on a micron scale, such as live cell cutting and injection [14]. Compared with traditional manual cutting methods, ultrasonic vibration micro-cutting has higher precision as well as efficiency, and can achieve a high-quality cutting effect [15]. Compared with laser micro-cutting, ultrasonic vibration devices are compatible with most biological micromanipulation platforms and can be used only by installing the corresponding device module, which greatly reduces the cost of use and is conducive to the popularization of ultrasonic surgical tools in clinical medicine [16].

In corneal trephination, different from common rigid trephination, exerting continuous pressure on the cornea, ultrasonic-vibration-assisted corneal trephination separates tissues based on the combined effects of strong instantaneous impact acceleration, microstreaming, and ultrasonic cavitation generated by the end effector during ultrasonic vibration [13]. This is because the head of an ultrasound knife will produce physiological effects when it comes into contact with human tissues, including mechanical effects, cavitation effects, and thermal effects [17]. Ultrasonic mechanical vibration can cut tissues with a high protein density and collagen fiber. The force applied to tissues by an ultrasound knife is discontinuous. Cavitation crushing can destroy low-density tissue and help protect adjacent tissue. The thermal effect is more prominent in cases of medium and high ultrasonic intensity as well as long working hours [18]. Because of the high frequency and low-amplitude vibration applied to the tool during the cutting process, the target tissue can easily reach its elastic limit under the action of ultrasound, which can be separated easily. There are many advantages of ultrasound cutting, such as fine cutting (a cutting accuracy of up to 1 μm), little trauma [19] (an influence range as small as 3 μm), a low temperature, no eschar, fewer complications, and fast wound healing [20].

Corneal trephination is a kind of delicate microsurgery. Especially in lamellar keratoplasty, the precision of corneal trephination is required to be as high as micron-level in order not to damage the endothelium and Descemet’s membrane [4,5].

In view of the limitations of manual, robotic, and laser trephination, as well as the advantages of ultrasonic cutting described above, we propose a corneal trephination method by introducing ultrasonic vibration cutting and have designed an ultrasound-assisted trephination system [21]. On this basis, we have also carried out several preliminary experiments to prove the effectiveness of the system.

## 2. Biomechanical Properties of the Cornea

### 2.1. Hyperelasticity and Viscoelasticity of the Cornea

Different from common biological soft tissues, the cornea has two significant characteristics: hyperelasticity and viscoelasticity (including creep, stress relaxation, and hysteresis) [22], as well as various anisotropic and nonlinear properties [23].

These complex biomechanical properties can be described effectively by a material model containing a Mooney–Rivlin hyperelastic and a Prony series viscoelastic [24]. Through uniaxial tensile tests of corneal spline [25], the Mooney–Rivlin hyperelastic constitutive model of the cornea may be described as follows [26]: (1)$$W=\sum_{i,j=0}^{N}C_{ij}{\left({\bar{I}}_{1}-3\right)}^{i}{\left({\bar{I}}_{2}-3\right)}^{j}+\sum_{i=1}^{N}\frac{1}{D_{i}}{\left(J-1\right)}^{2}$$
where *N* is the order of expression, *C_ij_* is the model coefficient reflecting the material’s own characteristics, and *D_i_* is the volume modulus reflecting the material’s compressibility.

The Prony series viscoelastic constitutive model of the cornea is established via stress relaxation tests, which may be described as follows [27]:(2)$$E(t)=\sum_{i=1}^{n}E_{i}e^{-t/\tau_{i}}+E_{n+1}$$
where *n* is the order of expression, *E_i_* is the relaxation modulus, and $\tau_{i}$ is the relaxation time.

As shown in Figure 1, corneal material models including Mooney–Rivlin and Prony series constitutive equations are simulated in ABAQUS. In Figure 1a, the relationship curves of corneal stress–strain show obvious linear features after a nonlinear rising stage. In Figure 1b, the behaviors of the cornea under a stress relaxation test are composed of an immediate stiff response, a transient relaxation phase, and a final steady-state stage [28].

Many researchers have studied the cornea as a whole, but they did not indicate a specific structural layer with regard to the measured biomechanical parameters. We can approximately consider that most of the testing results are biomechanical responses of the stroma according to corneal morphology and biomechanical properties [28,29].

### 2.2. Stratified Structure of the Cornea

The mean corneal thickness is 0.67 mm, and the central thickness is 0.55 mm. From a microscopic point of view the cornea has a complex layered structure [30], including the epithelium, Bowman’s membrane, the stroma, a novel pre-Descemet’s layer (Dua’s layer) [31], Descemet’s membrane, and the endothelium, from the outside to the inside, as shown in Figure 2.

In the layered structure of the cornea, the epithelium, the central stroma, and the endothelium are cellular layers. The composition of the three cell layers varies greatly. The stroma accounts for about 90% of the thickness of cornea, which is composed of 200–500 interwoven layers of collagen fibers embedded with proteoglycan. The endothelium and epithelium have higher in-plane stiffnesses compared to the stroma [30]. Bowman’s membrane, Dua’s layer [31] (the newly discovered pre-Descemet’s layer), and Descemet’s membrane are extracellular structures that are acellular and stronger. Therefore, based on the complex structure of the cornea and its hyperelastic/viscoelastic properties, the cutting mechanism of corneal tissue is significantly different from that of other biological tissues.

### 2.3. Mechanical Behaviors of the Interaction between Rigid Surgical Instruments and the Cornea

In order to simplify the mechanical behavior study of the interaction between rigid surgical instruments and the cornea, we took the process of a needle piercing the cornea as an example to measure the micro-force of a needle piercing the cornea. As shown in Figure 3a, the test system consists of a testing machine, fresh pig eyeball samples, different types of surgical suture needles, and a test bed; among these, we used an Instron 5848 Micro Tester as the testing machine, provided by Institute of Mechanics, Chinese Academy of Sciences. It has a load resolution of 50 mN and a position accuracy of 1 nm. The surgical suture needles were fixed on the test machine through a self-made needle handle gripper, which drove the suture needles to move in the vertical direction with a stroke of 100 mm and a speed of 0.3 to 25 mm/s. Figure 3b shows the corneal deformation during the whole piercing process. To simplify the problem, the cornea is roughly divided into three layers when describing corneal deformation: the upper surface, the stroma, and the lower surface. Figure 3c shows the relationship curve between the stress of the suture needles and the penetration displacement.

As shown in Figure 3b, the process of corneal puncturing by suture needles can be divided into two stages: the contact stage before the suture needle punctures the corneal epithelium (Figure 3b—1,2,3) and the traversing stage after the suture needle punctures the cornea (Figure 3b—4,5,6). The second stage can be further divided into two substages: the traversing stage of the needle tip after the upper surface is punctured (Figure 3b—4,5) and the traversing stage of the needle shaft after the lower surface is punctured (Figure 3b—6).

In Figure 3b—1,2,3, during the first stage the position of the initial piercing point on the upper surface of the cornea was taken as a reference (shown by the blue dashed lines in Figure 3b—1,2,3). With the vertical downward movement of the suture needle, the position of the corneal piercing point continued to move downward without puncturing (shown by the red dashed lines in Figure 3b—2) or was on the verge of being punctured (shown by the red dashed lines in Figure 3b—3). This indicates that the biomechanical properties of corneal materials that lead to the deformation of corneal materials do not change synchronously with changes in external forces, and that there is certain hysteresis behavior. Corneal tissue cannot be punctured if the cutting force does not exceed the corneal tissue cutting threshold. In contrast to Figure 3c, this stage corresponds to the contact of the suture needle on the corneal surface from point 0 to the puncture point, A. At this stage, because the corneal surface was not punctured, the needle axis force was completely manifested as the surface contact stiffness force.

In Figure 3b—4,5,6, during the second stage, i.e., after puncturing the corneal tissue, the position of the puncture point on the upper surface of the cornea is taken as the reference (shown by the blue dotted line in Figure 3b—4,5,6). In the first substage of the second stage, with the vertical downward movement of the suture needle, the position of the puncture point on the upper surface of the cornea firstly moves downward (shown by the red dotted line in Figure 3b—5). Compared with Figure 3c, this stage corresponds to stages B to C. Since the suture needle had only slightly penetrated into the cornea, the friction resistance was not obvious, and the cutting force generated by the two edges of the suture needle made the force obviously unstable.

During the second substage, i.e., after penetrating the lower surface of the cornea, the upper surface of the cornea showed an upward recovery motion at the entry point (red dashed line in Figure 3b—6). According to Figure 3c, this stage corresponds to the stage after point C, and the needle axis force is mainly manifested as friction resistance in addition to presenting a stable and linear upward trend. Due to the hyperelastic/viscoelastic properties of the corneal tissue, the deformation of the corneal tissue showed little recovery during the passage of the needle axis through the cornea after penetrating its lower surface. This indicates that, when the instrument exerts continuous pressure on the cornea, the actual corneal penetration depth has an obvious nonlinear relationship with the instrument feed displacement; once the cutting threshold is exceeded it is easy for the cutting depth to be excessive. This explains why manual and robotic trephines have inherent limitations.

## 3. Introduction of Ultrasound to Corneal Trephination

### 3.1. Feasibility and Advantages of Ultrasound-Assisted Corneal Trephination

As the force applied to tissues by an ultrasound knife is discontinuous, it is feasible and beneficial to introduce ultrasound into the field of corneal trephination based on the advantages of ultrasonic cutting as well as the applications of ultrasound in cell microcutting, as described in the Introduction. On this basis, we propose a conceptual diagram of ultrasound-assisted corneal trephination. As shown in Figure 4, in this conceptual diagram, unlike ordinary mechanical trephines, the end effector driven by ultrasound has a sharp end. We need to design a robotic system that can drive this end effector to move circularly and meet the specific requirements for completing corneal trephination operations.

### 3.2. Cutting Mechanism of Ultrasound-Assisted Corneal Trephination

Figure 5 shows a schematic diagram of ultrasound-assisted corneal trephination with a microscopic perspective. Cutting requires a relative displacement between the end effector and the cornea. There are two ways to generate this relative displacement: The first is that the biological tissue is fixed and the end effector moves along a predetermined cutting path. The second is that the end effector does not move macroscopically, except for ultrasonic vibration microscopically, and that the biological tissue moves along a predetermined cutting path. In order to simulate the operation of a trephine to a great extent, we chose the former. As shown in Figure 5, the end effector vibrates continuously in the vertical and tangential directions of the cutting surface.

As shown in Figure 5, during the trephination process the position of the cornea does not move, and the end effector vibrates continuously in the vertical direction while rotating in the *xoy* plane. The motion trajectory of the end effector in the *xoy* plane relative to the cornea is as follows:(3)$$\left\{\begin{array}{l} S_{x}(t)=v_{x}t\\ S_{y}(t)=v_{y}t \end{array}\right.$$
where *v_x_* and *v_y_* are the feed speeds of the end effector in x and y directions according to the cutting track, respectively. The vibration trajectory of the end effector along the z direction is as follows:(4)S(t)=Acos(2πft)
where *A* is the amplitude of vibration displacement and *f* is the vibration frequency.

The cutting speed of the end effector in the z direction relative to the cornea can be expressed as follows:(5)v(t)=−A⋅2πfsin(2πft)

In ultrasonic vibration cutting, the end effector first generates vibration and transfers the vibration energy to the part of particles directly in contact with it, and then the energy is gradually introduced into the tissue. The corneal tissue consists of several layers; each layer is regarded as being composed of many particles closely connected with each other. Once a particle in the medium is disturbed, the particle will have a movement away from its equilibrium position, which is bound to push its adjacent particles to move, such that each layer of particles will alternately be subjected to the pressure or tension caused by the wave. A particle here is actually a macroscopically small tissue volume element whose physical properties are approximately uniform. The small tissue volume ΔV has the mass *ρ*ΔV. In an ideal case, without considering energy attenuation, the particle adjacent to the end effector also follows the trajectory motion shown in Equation (4), and the particle’s vibration acceleration can be obtained through continuous derivation:(6)$$a(t)=-A\cdot {\left(2\pi f\right)}^{2}\cos(2\pi ft)$$

The force on the particle is as follows:(7)$$F=\Delta m\cdot a(t)=-\rho \Delta {\left(2\pi f\right)}^{2}\cos(2\pi ft)$$
where *ρ* is the density of the cornea.

As can be seen in Equation (7), when the ultrasonic cutting vibration frequency and acceleration are very high a great local alternating cutting force will be generated, which can easily reach the elastic limit of the cornea and cause it to break and be divided. As shown in Figure 6, according to relevant studies, if mechanical vibration with a particle acceleration of equal to or more than 5 × 10^4^ g is applied to living biological tissue then tissue can be quickly cut and will not hurt the surrounding tissue [32]. In order to meet the requirement of the cutting acceleration of corneal tissue, an ultrasound-assisted corneal trephination system with a working frequency of 35 kHz was designed.

## 4. Design of an Ultrasound-Assisted Corneal Trephination System

### 4.1. System Design Requirements Analysis

Corneal trephination is the first key step of corneal transplantations, requiring precise force and displacement control. During corneal trephination, on the one hand, excessive force can easily damage the iris and lens; on the other hand, the higher the precision of trephination the higher the success rate of surgery. In addition, in clinical practice doctors need to replace trephines with different diameters according to the size of the corneal lesions. On this basis, the design of an ultrasound-assisted corneal system should satisfy the following main requirements: (i) the end effector driven by ultrasonic vibration can realize corneal trephination movements with two degrees of freedom, including vertical motion and circular motion; (ii) the end effector can perform circular motions of different radii to satisfy the requirement of cutting corneal lesions with different diameters; (iii) the comprehensive configuration of the system should meet the accuracy requirements of the surgical operation; and (iv) it is safe and will not harm the patient or doctor during the operation.

### 4.2. Overview of System Structure Design

Based on the design requirements, high micro-dissection precision, and little trauma of ultrasound, discussed above, we designed the overall structure of a system, as shown in Figure 7.

The whole system includes off-the-shelf and custom-made equipment as well as electronics. Figure 7 shows a layout of its main components [21].

On this basis, as shown in Figure 8, we developed and manufactured an ultrasound-assisted corneal trephination system.

As shown in Figure 7 and Figure 8, there are four main components in the system, including an ultrasonic generator, a robot body, a system motion controller, and an upper computer. The ultrasonic generator and the corresponding end effector (ultrasonic scalpel) are the soul of the system, which can generate ultrasonic vibrations and perform the surgical operation, respectively. The robot body is the hardware mechanism that supports the end effector in completing the corneal trephination operation with different diameters. Through the control signal transmitted by the system motion controller, the robot can perform corresponding movements. The upper computer is the command sender of these control signals through Ethernet Bus. Based on the standard TCP/IP protocol instructions, the robot body can perform corresponding operations and the ultrasonic generator can also be controlled to turn on or off.

### 4.3. Design of the Ultrasound-Assisted End Effector

The ultrasound-assisted end effector is the core component of the system, which is the premise of system mechanism design. In order to meet the requirements of the cutting operation in corneal trephination, we firstly designed an ultrasound-assisted end effector (an ultrasonic scalpel) on the basis of 3D entity modeling and a modal analysis through simulation software, Pro/E (Figure 9).

As shown in Figure 9, the ultrasound-assisted end effector consists of three main parts: an ultrasonic transducer, an ultrasonic amplitude transformer, and a surgical blade. There are three design essentials of the end effector: (1) As the surgical blade is a consumable product, a groove has been designed at the end of the ultrasonic amplitude transformer to install the surgical blade, which can be clamped by bolts, so that the blade is convenient to replace. (2) In order to achieve the effect of simultaneous cutting along the corneal surface in vertical and tangential directions, according to the principle of ultrasonic wave propagation in solids, the longitudinal vibration transducer and the curved-shape surgical blade (type 12D, a double-edged blade) are combined to realize a two-dimensional elliptical ultrasonic vibration trajectory, which is made up of longitudinal and bending vibration directions of the tip of the blade, so as to obtain an optimal two-dimensional vibration mode for corneal trephination rather than only a single longitudinal trajectory in practical application. In consideration of the disadvantages of exerting continuous pressure on the cornea, as described in Section 2.3, a two-dimensional vibration mode can make not only the longitudinal force of the surgical blade on the cornea but also the tangential force along the surface of the cornea discontinuous in the process of corneal trephination, which can improve the surgical effect. Specifically speaking, ultrasonic vibration in a vertical direction, i.e., the longitudinal vibration direction of the tip of the blade, can guarantee a specific cutting depth in corneal trephination; ultrasonic vibration in a tangential direction of the corneal trephination trajectory, i.e., the bending vibration direction of the tip of the blade, can guarantee the cutting effect in a circular motion of the surgical blade. (3) As shown in Figure 9d, we used finite element analysis software, Pro/E, to simulate the motion trajectory of the surgical blade. The results show that the tip end can generate an elliptical vibration trajectory, which can meet the two-dimensional vibration requirements of corneal trephination.

### 4.4. System Mechanism Design

After the manufacture of the ultrasound-assisted end effector, we designed the motion actuator mechanism (robot body) of the end effector.

As shown in Figure 10, in order to meet the requirements of surgical operations, the robot body is mainly composed of the following four parts: (i) a precise two-axis motion platform with which to realize the feed motion of the end effector in a vertical direction and the circular cutting motion in the *xoy* plate (Figure 5) (including single-axis independent motion and two-axis linkage); (ii) an eccentric adjusting mechanism with which to realize the task of cutting the cornea lesions with different diameters without changing the surgical blade; (iii) the ultrasound-assisted end effector, as we discussed in the previous section, is designed to realize a two-dimensional elliptical ultrasonic vibration trajectory made up of longitudinal and bending vibration directions of the tip of the blade and to meet the cutting accuracy and incision smoothness requirements of corneal trephination; and (iv) an eyeball bracket and XY micro-displacement platform: The eyeball bracket is used to fix the surgical experimental object, and the XY micro-displacement platform is used to realize the center alignment of the surgical object and the ultrasonic scalpel. In order to realize the real-time detection of cutting force during surgery, there is a micro-force sensor (type 300G, Hyforcell Company, with a vertical force measuring accuracy of 1 mN, one DOF) embedded into the contact surface between the eyeball bracket and the XY micro-displacement platform, which can be seen in the section view of the eyeball bracket in Figure 10b.

(i)Design of the Precise Two-Axis Motion Platform

The platform can realize the feed motion of the end effector in the vertical direction and the circular cutting motion in the *xoy* plane (Figure 5).

First of all, the vertical feed motion along the vertical direction is realized by the ball–screw slider mechanism (type LCD02-10-2-L225-LC, Yiheda Company, Dongguan, China) driven by a vertical servo motor (type MSMF012L1V2M with brake, Panasonic Company, Kadoma, Japan), which can convert the rotational motion of the screw into the vertical translation motion of the slider. In this way, the ultrasonic scalpel can cut into or move up from the cornea surface conveniently. According to the design requirements, the platform needs to have a certain operating space and the advantage of a compact structure; therefore the maximum stroke of the Z axis of the platform is set as ±25 mm. The main technical points of vertical motion include the following:
(1)The vertical motion range of the experimental platform can be constrained by the vertical motion limit switch at both ends of the screw stroke, which can protect the platform from unexpected cases of motion that exceed the limit and complete the initialization operation of returning to zero of the vertical motion.(2)The scale grating (type RH100X30D05A, Renishaw Company, Wotton-under-Edge, UK) is installed on the side of the slide rail, and the closed-loop feedback control of the vertical feed motion can be realized by collecting the data of the grating scale; the depth closed-loop control accuracy can reach 1 μm.(3)Furthermore, in order to ensure surgical safety, a software limiter was also utilized in the robotic system control algorithm on the basis of adding limit switches (Figure 10a) to particular locations as necessary. Additionally, based on the real-time force data detected by the force sensor (with a data acquisition frequency of 500 KHz), cutting force feedback closed-loop control was performed to protect the cornea from excessive cutting (a detailed explanation can be seen in the next section (Section 4.5)). In this way, the reliability of the robotic system can be enhanced.

Secondly, the circular motion in the *xoy* plane (Figure 5) is realized by the eccentric adjustment mechanism driven by the rotary servo motor. As the eccentric position of the end effector can be adjusted by the eccentric adjustment mechanism, when the rotary servo motor rotates the blade tip of the end effector can move in a circle. The main technical points of rotational motion include the following:(1)The rotation zero is marked by the rotation zero sensor. The initialization operation of the zero return of the rotation degree of freedom can be completed automatically after each power-up of the platform.(2)A rotating conductive slip ring is introduced under the coupling of the rotating part, and the platform is connected with the eccentric adjusting mechanism through the slip ring, which can avoid the possible winding phenomenon of the power cord of the ultrasonic generator caused by rotating motion. Consequently, the adjustment range of the end effector’s rotational degrees of freedom is unlimited, and it can rotate forwardly as well as reversely in an infinite range.

In order to satisfy the operation requirements, the high-precision two-axis motion platform can simultaneously realize the feed motion in the vertical direction and the rotational motion around the axis direction (including single-axis independent motion and two-axis linkage).

(ii)Design of the Eccentric Adjusting Mechanism

To satisfy the requirement of cutting corneal lesions with different diameters, a modular eccentric adjustment mechanism has been designed [33]. The design details can be seen in Figure 11. The mechanism mainly consists of five parts: an eccentric adjusting plate, a micro-guide rail (type SSEB6-40, MISUMI), a micrometer head (type EPD02, Yiheda Company), a jacking block with a spring, and a rotary transition sleeve for assembling the ultrasonic scalpel. As shown in the side view (Figure 11b), there are two micro-guide rails, which are symmetrically distributed below the eccentric adjusting plate. The micro-rail sliders are fixedly assembled with the eccentric adjustment plate.

The eccentric adjustment function can be realized as follows: The measuring rod can be pulled back and forth when rotating the micrometer head, and thus the rotary transition sleeve can be moved back-and-forth along the micro-guide rail within 5 mm. Consequently, the corneal trephination diameter can be adjusted within 10 mm steplessly and accurately when the scalpel is driven to rotate by the rotating servo motor (type MSMF012L1U2M, Panasonic Company), so that the mechanism is suitable for corneal trephination tasks with different diameters of corneal lesions.

Additionally, the whole eccentric adjusting mechanism, driven by a servo motor (type MSMF012L1V2M with brake, Panasonic Company), can move along its vertical axis by utilizing lead screw–nut (type LCD02-10-2-L225-LC, Yiheda Company) transmission, as described in the vertical feed motion.

### 4.5. Human–Machine Interaction Interface Design

As shown in Figure 7 and Figure 8, we used a motion control card (type SMC304, Leadshine Company, Shenzhen, China) to establish the main control system. Based on the Ethernet communication protocol supported by the control card, we designed a human–machine interaction interface based on Labview. Through the human–machine interaction interface, operators can send control instructions to the mechanical system from the upper computer, and the feedback signals can also be sent back to the upper computer automatically.

Figure 12 is the front panel of the VI program of Labview; the human–machine interaction interface is divided into the following seven function modules: (a) A curve display module: the relationship curve between the force exerted on the cornea and the displacement of the vertical downward motion of the end effector can be displayed in real time. (b) A system status indicator module, which shows the operating status of the equipment through indicator lights. (c) A force and displacement data real-time acquisition module: real-time display of the detection data of the ruler grating and micro-force sensor through the numerical display control of Labview, including the movement displacement (the same is ideally the case with the cutting depth) and cutting force of the scalpel. (d) A closed-loop control module, which sets the limit pressure or displacement value according to different operations through numerical input controls. Once the system monitors the relevant measurement parameter values (including the cutting displacement and cutting force) to reach the threshold, the robotic system will automatically stop; that is, the system can be limited by software to ensure the safety of the operation process and the accuracy of the control system. (e) A vertical motor control module, which includes ① a motor signal display area to indicate the motor rotation direction and monitor whether the system reaches the hardware limit or zero point in real time through the indicator light; ② a motor parameters’ setting area to set the movement speed and range of the motor through numerical input controls; and ③ a motor control area to start up and shut down the motor through button controls. (f) A rotary motor control module, consisting of the same function areas of the vertical motor. (g) A main control module, consisting of ultrasonic power supply button control, the simultaneous movement of two-axis button control, and system exit button control. Additionally, in order to facilitate the recording and storage of data, on the basis of a real-time display of the waveform data-saving button control is added, and the specific data in the curve can be stored in an Excel file to evaluate the cutting effect.

## 5. Experiments of an Ultrasound-Assisted Corneal Trephination System

### 5.1. Human–Machine Cooperative Experimental Procedures

After the analysis of manual surgical procedures, a flowchart of human–machine cooperative operation procedures (Figure 13) has been determined. The surgery is divided into four stages according to the manual corneal trephination procedures [3]. Surgeons play a dominant part in the whole flowchart, including preparation work and then surgical robot control, as well as monitoring progress in real time. More specifically, in the first stage the doctor needed to use the XY micro-displacement platform below the eyeball bracket, shown in Figure 10a, to locate the center of the eyeball. Similarly to manual corneal trephination, the center-positioning operation is performed as long as the tip of the end effector in the zero position is aligned with the center of the eyeball pupil. Secondly, the corneal trephination diameter was set in the second stage using the eccentric adjustment mechanism shown in Figure 12. Thirdly, the detailed parameters, including the trephination diameter, cutting speed, moving displacement of the tool, and cutting angle, were set through the human–machine interface (Figure 12), and finally the robot control commands were sent to perform the surgical task. In addition, in future research, the configuration of the robot should be optimized to introduce a microscope into the operation.

### 5.2. Validity Verification of the Eccentric Adjusting Mechanism

In order to verify the precision control effect and eccentricity adjustment function of the system, an experiment of cutting grapes with the ultrasound-assisted end effector was designed. Concentric circles with different diameters were cut on the surfaces of grapes successfully without changing the surgical blade, and the motion control effect of the system was verified. The experimental flow chart is shown in Figure 14.

According to the experimental flow shown in Figure 14, the cutting operation of the ultrasound-assisted end effector on the surface of the grape was realized, and the cutting effect is shown in Figure 15.

The eccentric distance can be adjusted by manually rotating the micrometer head, which can realize the cutting operation of concentric circles with different diameters on the surfaces of the grapes without changing the working tool. It can be seen from Figure 15 that the concentric circular incisions are neatly arranged. The cutting diameter is adjustable, the cutting depth is precise and controllable, and the design effect of the platform has been achieved.

### 5.3. Comparative Experiments of Micro-Dissection with or without Ultrasound Assistance

As shown in Figure 16a,b, in order to verify the effect of the ultrasound-assisted surgical operation we performed ultrasound-assisted trephination experiments on grapes and porcine eye models in vitro with a frequency of ultrasonic vibration of 35 KHz and an amplitude of 10 μm. Under different experiment conditions, comparison curves of the relationship between the cutting force and displacement with and without ultrasonic vibration assistance were drawn.

The experiments were performed according to the flow chart shown in Figure 13. After the center-positioning process, the trephination diameter was set by rotating the micrometer head of the eccentric adjustment mechanism. The operation parameters (for example, a cutting depth of 0.5 mm, a vertical feed speed of 2 mm/s, a rotary cutting speed of 1 r/s, and a rotation angle of 360°) were then set through the human–machine interaction interface based on Labview. Finally, the operations can be completed when the instructions were sent from the upper computer and received by the robotic system.

It can be seen from Figure 16c,d that the trephination pressure increases with an increase in the displacement of the ultrasound-assisted end effector (the same is ideally the case with the cutting depth of the cornea), and the cutting force decreases rapidly after reaching the destruction threshold of the experimental subject. For the grape cutting experiment in Figure 16c, as the toughness of grape skin is the highest, the peak value of the cutting force corresponds to the moment when the grape skin is punctured. After the knife tip is inserted into the grape flesh, as water is the largest component of the grape flesh, the cutting force decreases rapidly. For the porcine cornea in vitro cutting experiment in Figure 16d, as lamellar keratoplasty is simulated here, the peak value of the cutting force corresponded to the moment when the corneal upper surface is punctured. As described in Section 2.2., the endothelium and epithelium have higher in-plane stiffnesses compared to the stroma [30]; the cutting force decreased rapidly after the knife tip punctured into the stroma. In both cases, once the surface of the cutting objects is penetrated the biological tissues cut by the tip of the surgical blade are all changed, so the force values with and without ultrasonic vibration all drop sharply after the peak. Generally speaking, by comparing Figure 16c,d, it is found that because the toughness of the corneal surface is much higher than that of grape skin, the cutting force in the porcine corneal cutting experiments with or without ultrasound are all larger than that of grape cutting. Specifically speaking, in each figure of Figure 16c,d, after ultrasound was applied, the cutting force in both corneal cutting and grape cutting was significantly reduced, which increased the reliability of the operation. It can be inferred that the peak (i.e., the “destruction threshold of the experimental subject” mentioned above) is the moment when the tissue is punctured, and its abscissa values (displacement of the ultrasound-assisted end effector) of the peaks in the comparison curves of the two groups is relatively small with ultrasonic assistance. However, due to the elastic effect of the cutting objects, when ultrasound is not applied the real tissue cutting depth is much smaller than the movement displacement of the end effector [12]. Therefore, it is verified that ultrasound-assisted trephination can not only reduce the cutting force but also effectively reduce the movement displacement (the same is ideally the case with the cutting depth of the cornea) of the tool that reaches the destruction threshold of the experimental subject.

### 5.4. Effect Evaluation of Ultrasound-Assisted Corneal Trephination

In allusion to the lamellar keratoplasty of the in vitro porcine eye model, the corneal tissue with a circle incision (in the experiment shown in Figure 16b) was sliced and observed by a light microscope after H&E staining, which was also compared with the section microscopic image of lamellar keratoplasty under the operation condition of manual trephine [9], as shown in Figure 17. It can be seen that the ultrasonic scalpel made a shallow incision due to the small set of the moving displacement of the tool; however, it can also be seen from Figure 17b that ultrasound-assisted trephination showed a more regular and less fluctuating edge shape, and the edge of the corneal slice was damaged to a lesser extent, which preliminarily proves the feasibility and superiority of ultrasonic technology applied in corneal trephination in addition to laying an experimental foundation for research on ultrasonic-assisted corneal tissue trephination.

## 6. Discussion

As the mean thickness of the cornea is 0.67 mm, while the central thickness is 0.55 mm, the moving distance in the experiment is set to be less than 3 mm. As shown in Figure 16, the moving displacement of the end effector with ultrasonic vibration assistance is still more than the trephination depth of the cornea. This phenomenon is also related to the insufficient intraocular pressure of the experimental porcine eyeball; however, due to the influence of the ultrasonic vibration parameters, there is still much room for improvement in the closing degree between the moving displacement of the tool and the cutting depth of the cornea for ultrasound-assisted corneal trephination of in vitro porcine eyeballs. The relationship between the parameters of excitation frequency, power, amplitude, cutting force, and cutting speed of the ultrasonic vibration system and the degree of corneal tissue deformation, incision depth (which can also be called cutting depth or trephining depth), and corneal tissue loss will be further discussed in the future to optimize the ultrasonic system parameters. Additionally, a miniaturized high-power multidimensional ultrasonic vibration system will be constructed by adopting guided wave transmission technology, finite element technology, micro-drive technology, and other means, aiming to facilitate the high-precision control and operation of corneal trephination.

## 7. Conclusions

On the basis of an analysis of the biomechanical properties of the cornea, we firstly explained why common manual and robotic trephines [9] are unable to accurately control the depth of corneal trephination. As ultrasonic vibration cutting has the advantages of high precision and little trauma, while being widely used in micro-cutting, we propose a method of ultrasound-assisted corneal trephination in this paper. It is an extension of research from our previous study and aims to solve the problem of uncontrollable corneal trephination depth in the research on robot-assisted corneal transplantation surgery [34]. Due to the discontinuous cutting pressure and locally concentrated ultrasonic energy, it is easier for the ultrasound-assisted end effector to reach the cutting threshold of corneal tissue, and the cornea is able to be trephined accurately with minimal deformation. On the basis of the overall design of the system architecture diagram, we first developed and manufactured an ultrasound-assisted end effector. In order to drive the end effector to perform specific corneal trephination operations, we designed a corresponding mechanic system and its human–machine interaction interface. Based on surgical requirements, we proposed a preliminary exploration of human–machine cooperative experimental procedures. According to the procedures, we performed several verification experiments on grape models and porcine eyeballs in vitro. The research results show the following: (1) the proposed system can realize the automatic and high-accuracy control of corneal trephination operations through human–machine interaction; (2) ultrasound-assisted trephination can not only reduce the cutting force during operation, but also effectively improve the closing degree between the movement displacement of the tool and the trephining depth of the cornea; and (3) ultrasonic-assisted trephination showed a more regular and less fluctuating shape of the cut edge, with the edge of the corneal slice being damaged to a lesser degree. The preliminary experiments show that ultrasound-assisted corneal trephination is feasible. In future research, on the one hand we will continue to optimize ultrasonic vibration parameters to improve trephination quality; on the other hand we will improve the robot configuration to be more suitable for clinical operation.

## 8. Limitations of the Study

In the first stage of the human–machine cooperative experimental procedures, the doctor needed to use the XY micro-displacement platform below the eyeball bracket shown in Figure 10a to locate the center of the eyeball. In reality, this is very difficult. How to locate the center of the eyeball precisely and efficiently is a problem to be solved urgently in the future, and image-aided location may be a solution. In addition, the system design looks similar to industrial equipment, which should be improved to be suitable for clinic use. Finally, trephination effectively assisted by ultrasound needs to be analyzed with different parameters of ultrasonic vibration in the future.

## Figures and Tables

**Figure 1 micromachines-14-00438-f001:**
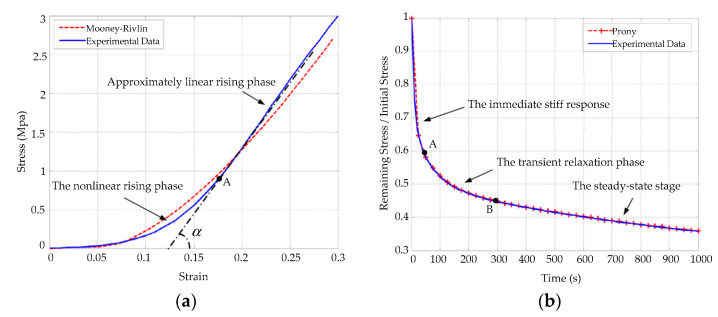
Simulation and experimental results of the corneal Mooney–Rivlin hyperelastic model and the Prony series viscoelastic model: (**a**) simulation and experimental comparison curves corresponding to Equation (1); (**b**) corneal stress relaxation curves corresponding to Equation (2). Simulation results are shown with dotted lines and experimental results are shown with solid lines.

**Figure 2 micromachines-14-00438-f002:**
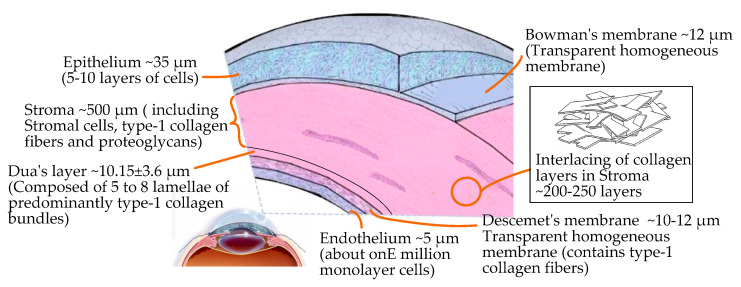
Microscopic layered structure and interior composition of the cornea.

**Figure 3 micromachines-14-00438-f003:**
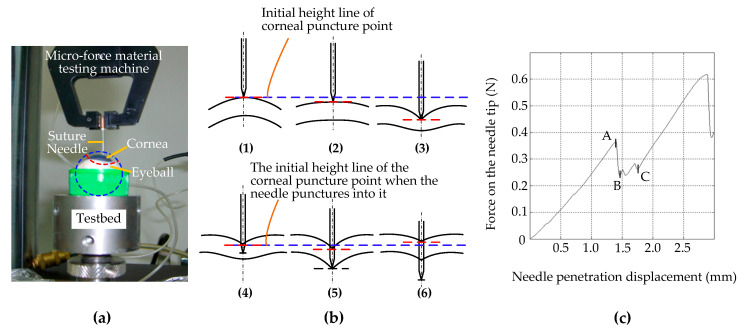
Cornea puncture experiment with suture needles. (**a**) Test system. (**b**) The corneal deformation during the whole piercing process: (1) The suture needle only just made contact with the upper surface of the cornea, and the force was 0. The corneal tissue was not deformed. (2) The needle began to exert force on the cornea, and the upper surface of the cornea was deformed but not punctured. (3) The needle continues to exert force on the cornea, close to puncturing the upper surface of the cornea. The degree of corneal tissue deformation increases gradually. (4) The suture needle punctured the upper surface of the cornea and began to penetrate the corneal stroma. (5) The suture needle continues to move down and is about to puncture the lower surface of the cornea. (6) The suture needle punctured the lower surface of the cornea and continued to puncture the inside of the eyeball, while the deformation of the corneal tissue showed a little recovery. (**c**) The relationship curve between the stress of the suture needles and the penetration displacement.

**Figure 4 micromachines-14-00438-f004:**
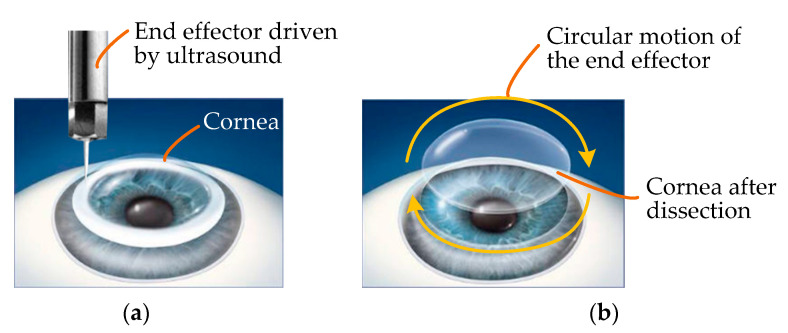
Conceptual diagram of ultrasound-assisted corneal trephination. (**a**) The cornea is cut by an end effector driven by ultrasound. (**b**) The end effector can move in a circle to simulate the operation of a trephine. In lamellar keratoplasty, the diseased part of the cornea can be removed by dissection when the cornea is trephined to a certain depth.

**Figure 5 micromachines-14-00438-f005:**
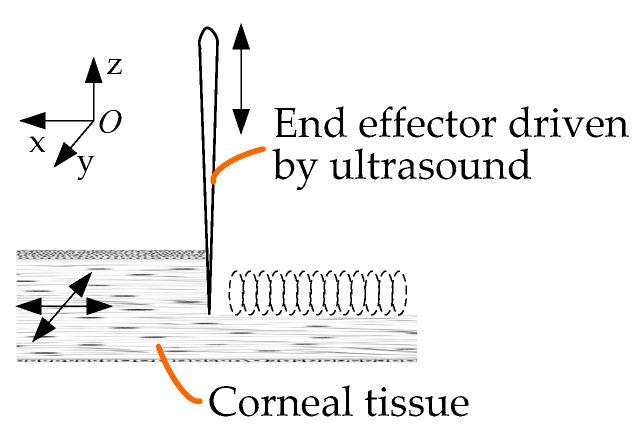
Schematic diagram of ultrasound-assisted corneal trephination with a microscopic perspective.

**Figure 6 micromachines-14-00438-f006:**
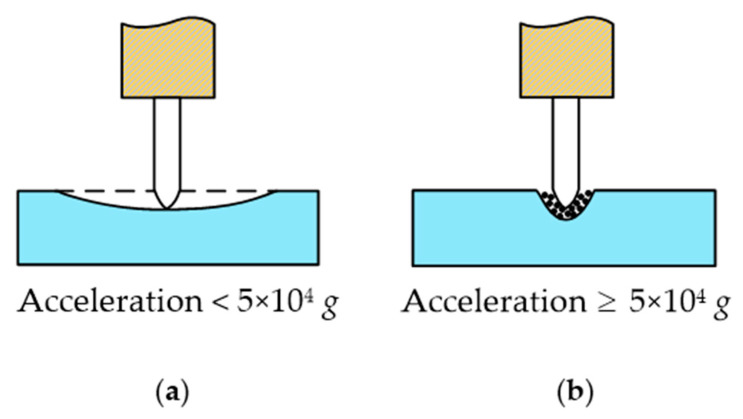
Different cases of the ultrasonic cutting of biological tissue. (**a**) If mechanical vibration with a particle acceleration of less than 5 × 10^4^ g is applied to living biological tissue then tissue cannot be cut, and the biological tissue would act in a similar manner to an elastic body with two fixed ends and the middle pressed down. (**b**) If mechanical vibration with a particle acceleration of equal to or more than 5 × 10^4^ g is applied to living biological tissue then tissue can be quickly cut and will not hurt the surrounding tissue.

**Figure 7 micromachines-14-00438-f007:**
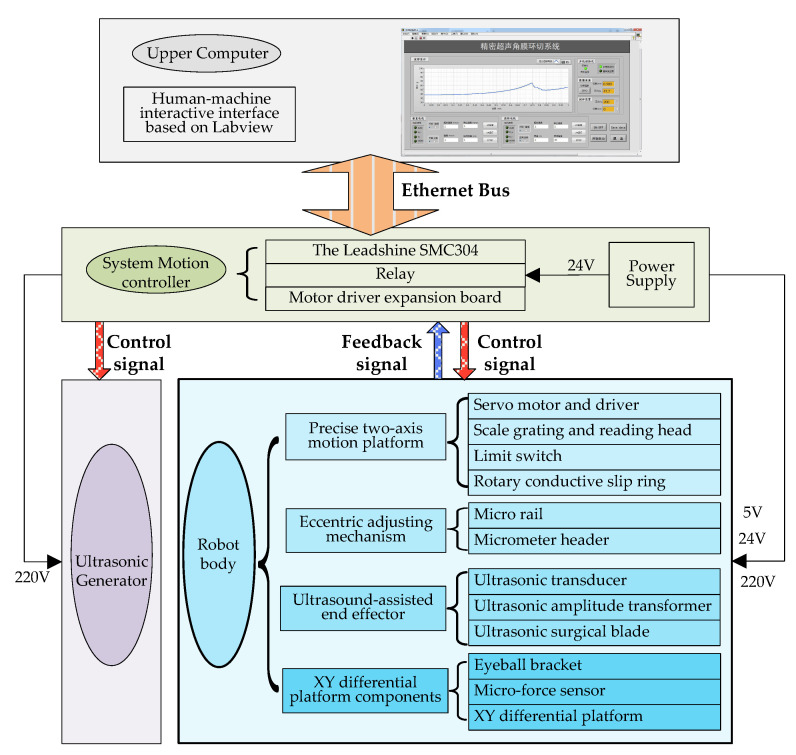
Overall system architecture diagram.

**Figure 8 micromachines-14-00438-f008:**
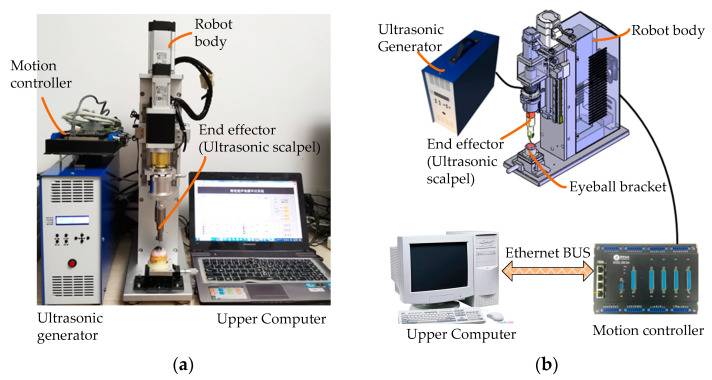
Physical map and component connection schematic diagram of an ultrasound-assisted corneal trephination system. (**a**) Physical map of the system. (**b**) System components’ connection schematic diagram.

**Figure 9 micromachines-14-00438-f009:**
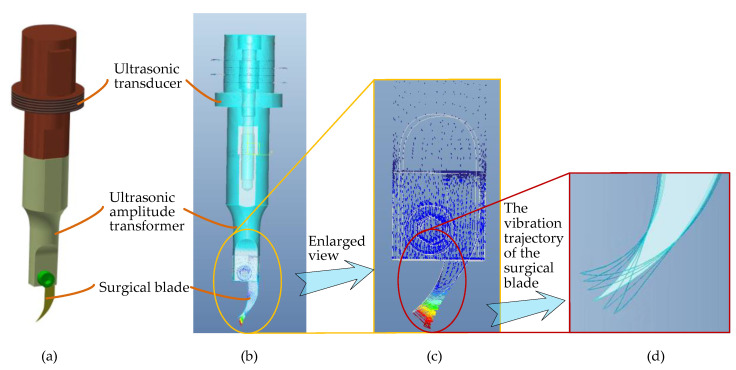
Three-dimensional entity modeling and a modal analysis of the end effector through simulation software, Pro/E. (**a**) Three-dimensional entity modeling of the end effector. (**b**) Modal analysis of the end effector. (**c**) Enlarged view of the surgical blade in the modal analysis. (**d**) Two-dimensional elliptical ultrasonic vibration trajectory of the tip of the surgical blade.

**Figure 10 micromachines-14-00438-f010:**
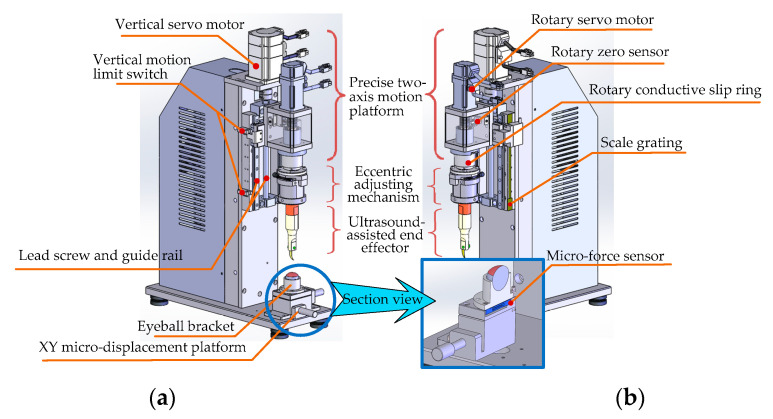
Structural detail of the robot body. (**a**) Left axonometric drawing. (**b**) Right axonometric drawing. Section view of the eyeball bracket and XY micro-displacement platform.

**Figure 11 micromachines-14-00438-f011:**
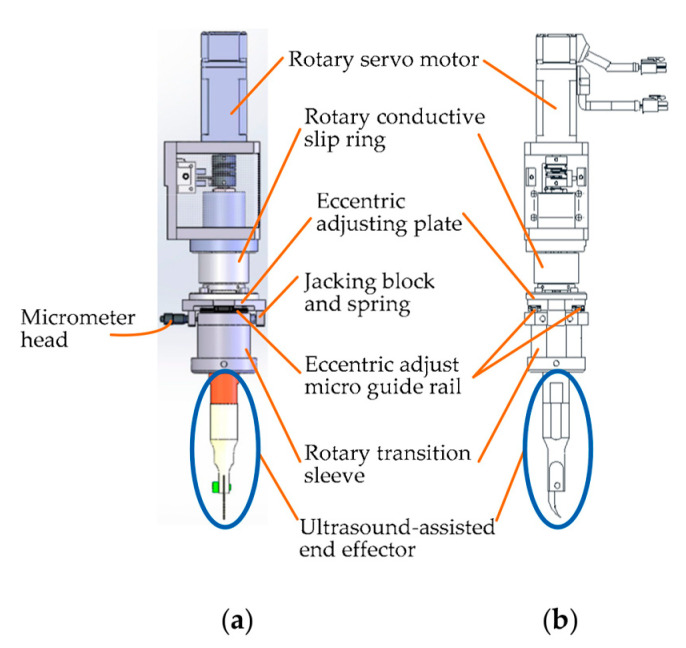
Design details of the eccentric adjusting mechanism. (**a**) Front view. (**b**) Side view.

**Figure 12 micromachines-14-00438-f012:**
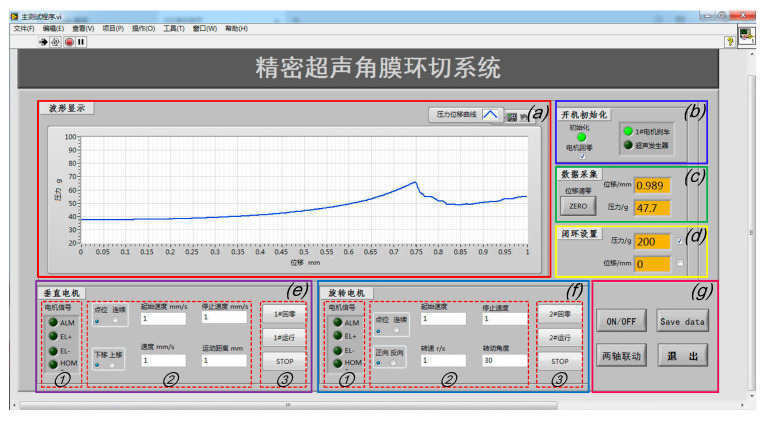
The human–machine interaction interface of the system based on Labview: (**a**) the cutting force–displacement curve real-time display zone; (**b**) system status display zone; (**c**) force and displacement data real-time display zone; (**d**) closed-loop setting zone; (**e**) vertical and (**f**) rotary motor control zone; and (**g**) the main control button zone.

**Figure 13 micromachines-14-00438-f013:**
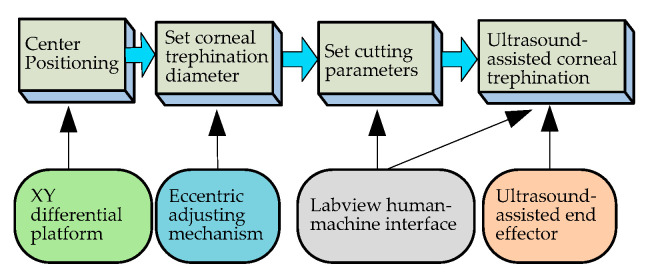
A preliminary exploration of human–machine cooperative experimental procedures.

**Figure 14 micromachines-14-00438-f014:**
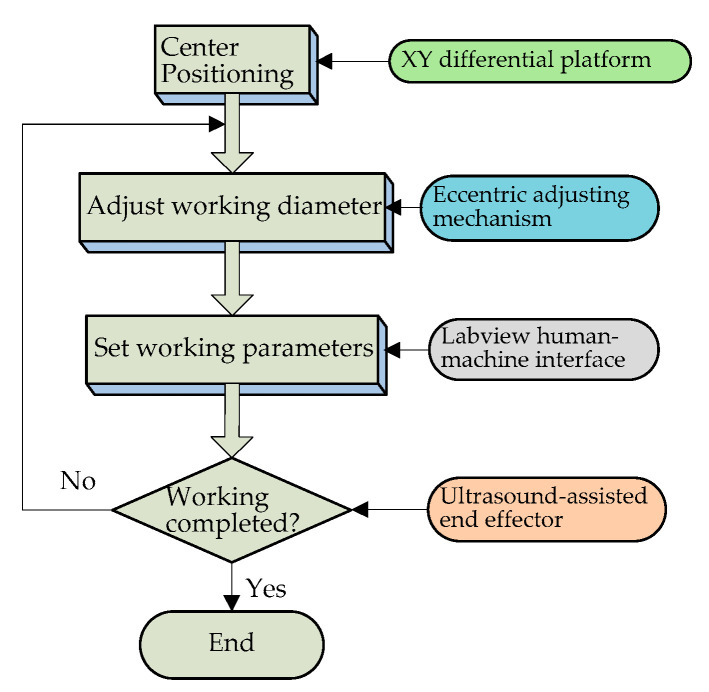
Experimental flow chart.

**Figure 15 micromachines-14-00438-f015:**
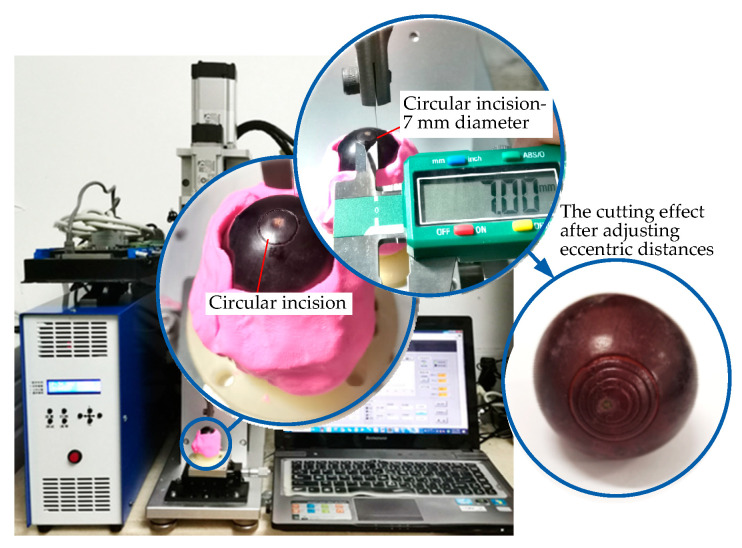
Validity verification experiment of the eccentric adjusting mechanism.

**Figure 16 micromachines-14-00438-f016:**
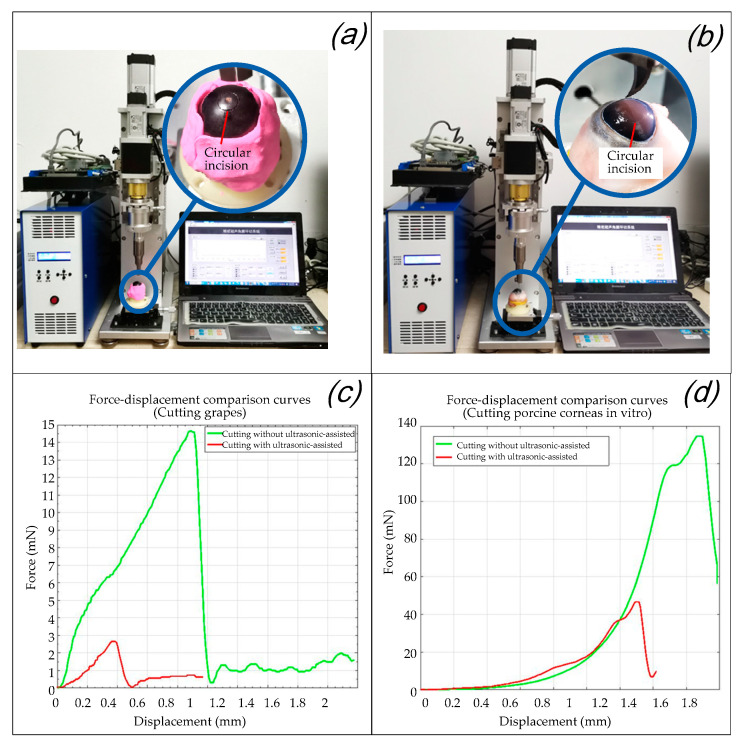
Validity verification experiment of the eccentric adjusting mechanism. (**a**) Trephination experiments on grapes. (**b**) Trephination experiments on porcine corneas. (**c**) Force–displacement comparison curves in the experiments of cutting grapes. (**d**) Force–displacement comparison curves in the experiments of cutting porcine corneas in vitro.

**Figure 17 micromachines-14-00438-f017:**
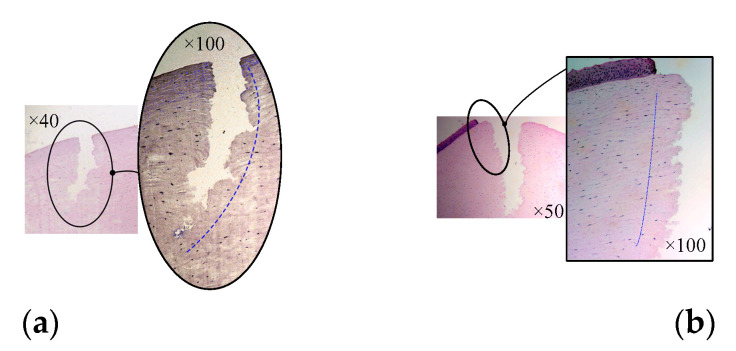
Microscopic images of corneal sections. (**a**) The cornea is cut by manual trephine. (**b**) The cornea is cut by the ultrasound-assisted end effector.

## Data Availability

The data presented in this study are available on request from the corresponding author. The data are not publicly available due to privacy.

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
