# Peer review of "Design and Experiment of an Ultrasound-Assisted Corneal Trephination System"

_micromachines, 2023, doi:10.3390/mi14020438_

Round 1
Reviewer 1 Report
This paper presents the design and test of an in-house developed ultrasound-assisted corneal trephination system. In general, this paper is well-orgainzed. However, the following issues need to be addressed:
1. This paper is too long. This paper seems an overview or report of the developed corneal trephination system, not a scientific research paper. It is required to reduce the length of the paper and focus on the main contribution.
2. It is suggested to move the contexts on the background to the introduction section, such as the first paragraph in Section 3.1.
Reviewer 2 Report
This is an article that combines ultrasonic vibration processing technology with medical technology. The authors have developed a mathematical model, simulated an ultrasonic vibration transducer, and performed related experiments. We think it is a good article, but there are still some problems. We think the article can be considered for acceptance after minor revision.
1. Figure 9(c) in the paper does not show the obvious elliptical amplitude trajectory, and we hope the authors can improve the figure.
2. The ultrasonic cutting blades shown in Figures 9 and 10 in the paper are curved, and the end of this shape of tool will show a two-dimensional elliptical vibration trajectory in practice. What is the basis for the authors' opinion that the elliptical vibration trajectory is optimal in corneal transplantation. What is the difference between elliptical vibration trajectory and single longitudinal trajectory in practical application, please elaborate.
3. In section 5 of the paper, the authors introduced the operating frequency and operating amplitude of the ultrasonic vibration transducer used in the experiment, and we believe that the exact frequency as well as amplitude values should be given in the experiment, rather than a range.
4. We believe that the experimental results shown in Figures 18(c) and 18(d) in the paper should be explained in detail, such as why the force values with and without ultrasonic vibration drop sharply after the peak, why the peak force is different, and how the above phenomenon affects the rupture phenomenon of the eye.
